# Transcriptome Profiling and Metagenomic Analysis Help to Elucidate Interactions in an Inflammation-Associated Cancer Mouse Model

**DOI:** 10.3390/cancers13153683

**Published:** 2021-07-22

**Authors:** Kazuko Sakai, Marco A. De Velasco, Yurie Kura, Kazuto Nishio

**Affiliations:** Department of Genome Biology, Kindai University Faculty of Medicine, Osaka 589-8511, Japan; kasakai@med.kindai.ac.jp (K.S.); mdev@med.kindai.ac.jp (M.A.D.); y.kurakindai@med.kindai.ac.jp (Y.K.)

**Keywords:** 16S rDNA gene sequencing, Piphillin, inflammation-associated cancer, PI3K-Akt-mTOR

## Abstract

**Simple Summary:**

Colitis-associated colorectal cancer is the third most significant condition that increases the overall risk of developing colorectal cancer. In this study, we examined normal colonic mucosa of tumor-bearing mice in the DSS/AOM mouse model by gene expression profiling and fecal samples by 16s rDNA amplicon sequencing. Gene set enrichment analysis revealed that genes associated with fatty acid metabolism, oxidative phosphorylation, and the PI3K-Akt-mTOR pathways were enriched colonic mucosa of DSS/AOM mice. Additionally, enrichment of the sphingolipid signal and lipoarabinomannan biosynthetic pathways were inferred from fecal microbial composition. Our findings provide insights into altered transcriptome and microbiome in a mouse model of colitis-induced carcinogenesis.

**Abstract:**

Colitis is a risk factor for colorectal cancer (CRC) and can change the dynamics of gut microbiota, leading to dysbiosis and contributing to carcinogenesis. The functional interactions between colitis-associated CRC and microbiota remain unknown. In this study, colitis and CRC were induced in BALB/c mice by the administration of dextran sodium sulfate (DSS) and/or azoxymethane (AOM). Whole transcriptome profiling of normal colon was then performed, and gene set enrichment analysis (GSEA) revealed enriched fatty acid metabolism, oxidative phosphorylation, and PI3K-Akt-mTOR signaling in the tissues from DSS/AOM mice. Additionally, immunohistochemical staining showed increased expression levels of phosphorylated S6 ribosomal protein, a downstream target of the PI3K-Akt-mTOR pathway in the inflamed mucosa of DSS/AOM mice. Fecal microbes were characterized using 16S rDNA gene sequencing. Redundancy analysis demonstrated a significant dissimilarity between the DSS/AOM group and the others. Functional analysis inferred from microbial composition showed enrichments of the sphingolipid signal and lipoarabinomannan biosynthetic pathways. This study provides additional insights into alterations associated with DSS/AOM-induced colitis and associates PI3K-Akt-mTOR, sphingolipid-signaling and lipoarabinomannan biosynthetic pathways in mouse DSS/AOM-induced colitis.

## 1. Introduction

Inflammatory bowel disease (IBD), which includes Crohn’s disease and ulcerative colitis, is an important public health problem [1,2]. Patients with IBD typically suffer from abdominal pain, diarrhea, weight loss, and rectal bleeding [2]. Moreover, patients with inflammatory bowel disease are at an elevated risk for developing colon cancer [3,4,5], which is dependent on the duration, extent, and severity of inflammatory disease [6,7]. The colonic mucosa of patients suffering with active disease shows infiltration of mononuclear and polymorphonuclear leukocytes, epithelial cell erosion, crypt abscesses, and epithelial cell hyperplasia [8]. Colon carcinogenesis is a complex process that involves various interactive responses from gut mucosa, stromal cells, including immune cells and fibroblasts, and intestinal microflora, which can be further influenced by host genetic susceptibility factors and environmental elements. Immune cells in the colon have emerged as the principal effectors in the pathogenesis of IBD and colitis-associated cancer (CAC) [9,10,11].

Animal models that recapitulate human diseases and have contributed greatly to the understanding of IBD-associated colon cancer. The dextran sodium sulphate/azoxymethane (DSS/AOM)-induced colon tumorigenesis model is the most widely accepted mouse model [12,13,14]. In this model system, DSS-mediated injury induces intestinal inflammation that contributes to colon carcinogenesis caused by AOM. This model is also useful in evaluating the efficacy of prolonged prophylactic or therapeutic treatment of colitis and colon cancer.

A large number of reports have confirmed the role of bacterial flora in the pathogenesis of colorectal cancer, outlining the structural composition of gut microbiota in patients with colorectal cancer [15]. Previous studies have found that there was a reduction in the abundance of colonic mucosal flora in patients with colorectal cancer, and an increase in Bacteroides was postulated to be related to the pathogenesis of colorectal cancer [3,4,16,17]. However, the relationships between gut microbes and colorectal carcinogenesis remain unclear. We hypothesized that the colonic microbes could influence colitis-associated cancer, therefore, we utilized a mouse model of colon carcinogenesis induced by AOM and DSS as a colitis-associated cancer model to test our hypothesis [18]. For this, we performed a comprehensive analysis using gene expression profiling on mucosal colon tissues and 16S rDNA sequencing in fecal samples and elucidated the interactions between gut microbes and inflammation in the host colon during carcinogenesis.

## 2. Materials and Methods

### 2.1. DSS/AOM-Induced Enteritis/Carcinogenesis Mouse Model

Female BALB/c mice were purchased from CLEA Japan (Tokyo, Japan). The animals were housed in the Animal Laboratory at Kindai University Faculty of Medicine. The mice (8–10 weeks of age) were maintained in a specific pathogen-free vivarium at 18 °C to 23 °C with humidity at 50% and with a 12-h/12-h dark/light cycle. The animals had free access to drinking water and a pelleted basal diet. AOM and DSS were purchased from Sigma-Aldrich (St. Louis, MO, USA) and MP Biomedicals (Santa Ana, CA, USA). All other chemicals used in this study were of reagent grade and were purchased from general laboratory suppliers. For the carcinogenesis study, mice were randomized into untreated control (*n* = 8), DSS (*n* = 8), AOM (*n* = 8), and DSS/AOM (*n* = 6) treatment groups. To induce CRC, mice received triple injections of 10 mg/kg AOM and/or 3 subsequent cycles of 2% DSS-supplemented water over a period of 4 weeks. Mice were monitored weekly for the presence of bloody stools and or diarrhea as indicators for the formation of colonic tumors. After 6 months, the mice were euthanized, and the colon tissue and fecal contents were collected for pathological examination, whole transcriptomic profiling, and 16S rDNA amplicon sequencing. The distal portion of the colon tissue that was macroscopically normal was fixed in 70% ethanol, paraffin embedded, sectioned and stained with hematoxylin and eosin (HE). The Institutional Animal Care Committee of the Kindai University Faculty of Medicine approved all the animal experiments (protocols KAME-30-003 (approval date: 30 January 2019) and KAME-2020-026 (approval date: 26 May 2020)).

### 2.2. Ion Torrent Sequencing Using the AmpliSeq Transcriptome Gene Expression Assay and Transcriptome Analysis

Total RNA was extracted from the mouse colon tissues using the AllPrep DNA/RNA Mini Kit, according to the manufacturer’s instructions (Qiagen, Valencia, CA, USA). The quality and quantity of the RNA were verified using the NanoDrop 2000 device (Thermo Fisher Scientific, Foster City, CA, USA) and the RiboGreen RNA assay kit (Life Technologies, Carlsbad, CA, USA). Whole transcriptome analysis of colon mucosal tissues from untreated control (*n* = 4), DSS (*n* = 4), AOM (*n* = 4), and DSS/AOM-treated (*n* = 6) mice was performed using the Ion AmpliSeq Transcriptome Mouse Gene Expression assay (Thermo Fisher Scientific) [16]. Ten nanograms of RNA were subjected to reverse transcription, followed by library preparation. The Ion Xpress Barcode Adapters were used for sample identification. Purified libraries were pooled and sequenced on an Ion Torrent S5 system. Of the 18 samples, 1 sample from the DSS group was excluded from the analysis because of its low sequence quality with a total read count of 2,652,660 and an on-target alignment rate of 38.2%. The average total number of reads for the samples used in the analysis was 9,639,739 (6,033,709–16,119,594), and the average on-target alignment rate was 52.0% (47.1–61.8%). The generated gene expression data were normalized, and low-expression genes were filtered out.

### 2.3. Gene Selection, Pathway Analysis and Gene Set Enrichment Analysis (GSEA)

Differentially expressed genes against DSS/AOM-treated samples were selected using log2 fold change (α = 0.01 and null distribution with 20 permutations) as cutoff values in Orange data mining software version 3.28 (https://orangedatamining.com/ access date: 20 March 2021) with the bioinformatics add-on version 4.3.1 [19]. Unsupervised hierarchical clustering was performed using Euclidean distance and Ward’s linkage, and clustered heatmaps were generated in Orange analysis software. To explore the potential biological pathways underlying carcinogenesis, differentially expressed genes were submitted to Enrichr and analyzed against reference pathways in the WikiPathways Mouse 2019 biological pathway database [20,21] Gene Set Enrichment Analysis (GSEA) [22,23] was performed to identify pathways enriched in the Molecular Signatures Database (MSigDB) Hallmark gene set. A normalized *p* value of <0.05 and an FDR (false discovery rate) *q* value of <0.25 were considered statistically significant. 

### 2.4. 16S Metagenomic Sequencing and Analysis

DNA was extracted from the fecal samples and bacterial 16S rDNA regions were amplified using PCR and two primer pools covering variable regions (V2–V4, V6–V9) from the Ion 16S Metagenomics Kit (Thermo Fisher Scientific), according to the manufacturer’s instructions [24]. For library preparation, the V2, V3, V4, V6, V7, V8, and V9 16S rRNA regions were amplified, followed by end-repair and barcoded-adaptor ligation using the Ion Plus Fragment Library Kit (Thermo Fisher Scientific). The pooled library was sequenced as single-end 400-bp reads using the Ion S5 sequencing kit (Thermo Fisher Scientific). The FASTQ files were analyzed using the CLC Genomics Workbench version 12.0 (Qiagen) and the Microbial Genomics Module (Qiagen). Sequence reads were clustered into operational taxonomic units (OTUs) with a 99% identify threshold against the Greengenes database, version 13.8. A set of sequences representing OTUs were analyzed using Calypso (version 8.84) [25]. OTU abundance was normalized with cumulative-sum scaling (CSS) and log2 transformation. Samples with a total read count <1000 were filtered from subsequent analyses. An analysis of similarities (ANOSIM) and a principal component analysis were carried out to assess dissimilarity. Redundancy analysis (RDA) was used as a constrained ordination method to determine the degree of variance according to variables [25,26]. Hierarchical clustering and heatmap, using the Bray-Curtis distance metric, and Linear discriminant analysis Effect Size (LEfSe) were carried out in Calypso. Pearson correlation analysis of OTUs correlated to treatment were generated using MicrobiomeAnalyst [27]. Prediction of Kyoto Encyclopedia of Genes and Genomes (KEGG) pathways in the control and DSS/AOM-exposed groups were inferred from OTU abundances in Piphillin [28].

### 2.5. Immunohistochemical Staining

A standard immunohistochemistry analysis was performed for the phosphorylated S6 ribosomal protein (p-S6), a downstream molecule of the phosphoinositide 3-kinase (PI3K)-protein kinase B (Akt)-mammalian target of rapamycin (mTOR) signal pathway. Ethanol-fixed tissue sections were sectioned and placed on positively charged slides and pretreated with steam heating in DAKO target retrieval solution (Catalog #S1699, Agilent, Santa Clara, CA, USA) for 20 min. Slides were incubated with anti-p-S6 (Ser 235/236) antibody (Catalog #2211, Cell Signaling, Danvers, MA, USA, dilution 1:100), overnight stained using the ABC kit (Vector Laboratories, Burlingame, CA, USA) following the manufacturer’s protocols, developed in diaminobenzidine DAB (Invitrogen), and counterstained with hematoxylin. Assessment of immunohistochemical staining was performed on scanned colon mucosa sections captured at 10x magnification. H scores were calculated for p-S6 positivity in the epithelial and stromal regions of the distal colon using QuPath v0.2.0-m4 image analysis software (https://qupath.github.io/ access date: 1 February 2021).

## 3. Results

### 3.1. Evaluation of the DSS/AOM-Induced Colon Cancer Model

To study the relationships associated between colon inflammation and colon carcinogenesis, we used the well-established DSS/AOM-induced colon cancer protocol to induce colon tumor formation in 8-10-week-old female BALB/c mice. Experimental groups consisted of untreated control, DSS-, AOM- and DSS/AOM-treated mice. In our study, all mice treated with the combination of DSS and AOM developed tumors after 6 months. No changes in mouse bodyweights were noted in the early phase of the experiment, however, mice in the DSS/AOM group experienced gradual bodyweight declines from day 78–112, which remained stable for the duration of the experiment (Appendix A). Overall, mice developed an average of 4.16 tumors ranging from one to 12 in the distal colon with an average maximum diameter of 4.92 mm (range 0.92–11.28 mm, Figure 1A). Histological evaluation confirmed the development of adenocarcinoma in all mice. Further histological evaluation of normal mucosa and normal-appearing mucosa adjacent to tumors showed features consistent with inflammation, including thickening of the submucosa accompanied with leukocyte infiltration in the submucosa and lamina propria in colons from DSS- and DSS/AOM-treated but not AOM-only treated mice (Figure 1B). Additionally, multiple lymphoid aggregates were found throughout the colons of DSS-, AOM-, and DSS/AOM-treated mice. The lack of tumor formation in AOM-only-treated mice indicated that induction of inflammation is required for colon carcinogenesis in mice, thus establishing the development of inflammation induced colon cancer. 

### 3.2. Whole Transcriptome Profiling

Having established inflammation induced carcinogenesis in our mouse model, we subsequently performed gene expression profiling of normal mucosa from the distal colons of untreated control, DSS-, and AOM-treated colons and normal appearing mucosa adjacent to tumors in DSS/AOM-treated mice. Gene expression data for mouse colon mucosal tissues were obtained using whole transcriptome analysis. After normalization, low-expression genes were filtered and a total of 256 genes differentially expressed in DSS/AOM-treated mice were selected for further profiling analysis. Hierarchical clustering analysis revealed clusters of genes upregulated or downregulated in DSS/AOM-treated mucosal samples (Figure 2). Normal mucosal samples from DSS- and AOM-treated mice clustered together and showed similar expression profile distinct from that of untreated controls and DSS/AOM-treated mice. Genes that were overexpressed in this cluster were downregulated in both DSS/AOM and control samples. Functional profiling showed that genes in this cluster were associated with peptide and class A G protein-coupled receptors (GPCRs), type II interferon signaling, chemokine-signaling pathways and macrophage markers (Figure 3A). Genes that were overexpressed in DSS/AOM-treated mice were downregulated in all other groups and were associated with the focal adhesion-PI3K-Akt-mTOR-signaling pathway, bone morphogenetic protein (BMP) pathway, class B GPCR, one carbon metabolism and inflammatory response pathway (Figure 3B). 

To further explore the functional differences of enriched genes sets in inflammation induced colon cancer, we evaluated differences in enrichment profiles of hallmark gene sets from the MSigDB database in DSS/AOM samples against untreated controls using GSEA. Notably, the top five enriched gene sets in the DSS/AOM group were protein secretion, mammalian target of rapamycin complex 1 (MTORC1) signaling, G2M checkpoint, downregulation of ultraviolet (UV) response and hypoxia (Table 1). These results provide preclinical evidence that link alterations of these pathways to the development of inflammation induced colon tumors.

### 3.3. 16S Metagenomic Sequencing

Next, we sought to determine whether differences of gut microbial composition could be correlated to DSS/AOM-induced colon carcinogenesis. For this, we performed 16S rDNA sequencing to analyze differences in microbial composition of fecal samples of untreated control, DSS-, AOM- and DSS/AOM-treated mice. We used proximal and distal fecal samples for these analyses. Rarefaction analysis indicated that our sequencing depth was adequate (data not shown). Overall significant differences in microbial composition were observed between the different treatments (Anoisim Bray-Curtis R = 0.499, *p* = 0.001). Moreover, site did not contribute to differences in microbial composition, however, DSS/AOM treatment accounted from maximal variation. (Figure 4A). We performed redundancy analysis (RDA) to evaluate the effect of treatment as a contributing factor influencing microbial composition. This analysis showed that treatment of DSS/AOM and the presence of cancer were indeed factors contributing to differences in microbial composition (Figure 4B). No significance was observed when the RDA was applied to site (RDA significance, *p* = 0.249). Subsequently, we performed species-level unsupervised hierarchical clustering of the top 50 OTUs with the highest abundance, to identify key factors associated with treatments. Overall, fecal samples from DSS/AOM-exposed mice showed distinct clustering from all other groups (Figure 4C). These data indicate that the presence of tumors is associated with fecal bacterial composition. 

Next, we aimed to identify taxa most likely to be associated with DSS/AOM-induced carcinogenesis. LEfSe analysis revealed genus-level taxa associated with fecal samples from tumor bearing DSS/AOM-treated mice compared to all other treatment groups (Figure 4D). We subsequently performed correlation analysis using Pearson’s coefficient to correlate individual OTUs to specific treatment. Of the top 25 OTUs, 19 were correlated to DSS/AOM samples, notably 11 of these (59%) were classified as *Clostridiales* and three (27%) were classified as *Staphylococcus aureus* (Figure 4E).

### 3.4. Functional Prediction Analysis Comparing Inflammation Induced Colon Cancer and Normal Mucosa

Bacterial communities tend to be resistant to stresses and show functional redundancy. To determine the effect of this phenomenon in our model of colon-induced carcinogenesis, we performed functional prediction analysis by community profiling in fecal samples from DSS/AOM-treated mice and compared it to samples from healthy untreated control mice. Functional predictions were inferred from the analysis of OTU abundance and was performed using the Piphillin platform to detect enriched KEGG pathways in fecal microbes. From this analysis, we extracted twelve pathways enriched in the DSS/AOM-exposed group (Figure 5A). Notable among these pathways were steroid synthesis, the lipoarabinomannan biosynthetic pathway and the sphingolipid-signaling pathway. These microbial metabolic pathways can influence host regulation of hypoxia, fatty acid metabolism, oxidative phosphorylation and PI3K/Akt signaling, which were all enriched in DSS/AOM mice (Figure 5B) [29,30,31,32]. Collectively, our data provides evidence to support the notion that the early inflammatory and carcinogenic effects of DSS and AOM treatments drive carcinogenesis and induce dysbiosis by altering gut microbial structure, which, in turn, may act to further promote cancer progression through bacteria-host interactions. 

### 3.5. Immunohistochemical Staining of Phosphorylated-S6 Ribosomal Protein

GSEA analyses revealed enrichment of genes associated with the PI3K-Akt-mTOR pathway. To confirm the finding, we examined activation of the PI3K-Akt-mTOR signaling pathway in the distal colonic mucosa by measuring the expression levels of phosphorylated S6 ribosomal (p-S6) protein, a downstream molecule of the PI3K-Akt-mTOR signaling pathway, using quantitative immunohistochemistry. We examined expression levels in the epithelial and stromal compartments of the distal colons of control, DSS, AOM and non-cancer regions of DSS/AOM mice. Levels of p-S6 were significantly higher in both the epithelium and stroma of the DSS/AOM group compared with other groups (Figure 6). The drastic increase in p-S6 in the colonic mucosa of DSS/AOM-treated mice is in agreement with our previous result and confirms DSS/AOM-induced induction of the PI3K-Akt-mTOR pathway at the functional level. 

## 4. Discussion

Several studies have shown that chronic inflammation in organs can greatly increase the risk of developing cancer [3,4,5]. An inflammatory component is also present in the microenvironments of tumors, and recent studies have begun to unravel molecular pathways linking inflammation and cancer [33,34]. In the tumor microenvironment, inflammation contributes to the proliferation and survival of malignant cells, angiogenesis, metastasis, the subversion of adaptive immunity, and reduced responses to hormones and chemotherapeutic agents. Recent data also suggest that an additional mechanism involved in cancer-related inflammation is the induction of genetic instability perpetrated by inflammatory mediators [35]. While natural flora is essential for maintaining homeostasis in a healthy immune system, opportunistic bacteria hijack commensal bacteria, promote dysbiosis and weakening the immune system. While a bounty of information is being gained on how bacterial dysbiosis influences cancer, much remains unknown due to the complex nature of the disease, its heterogeneity and context-specific characteristics. Ultimately, how bacterial flora affects host tissues functionally remains unclear. 

Here, we have shown that inflammatory and carcinogenic effects of DSS and AOM cooperate to induce carcinogenesis in mice. Moreover, our gene expression and metagenomic studies revealed altered transcriptomic signatures in tumor normal adjacent colonic mucosa and dysbiosis of fecal microbes. While our study is exploratory and a causal relationship has not been established, our findings suggest that structural compositional changes of gut microbes promulgated by the development colonic tumors may have further altered the normal adjacent mucosa, possibly increasing the risk of malignant transformation or further promoting cancer progression of established tumors. Changes in microbial composition, such as the *Firmicutes/Bacteroides* ratio, termed dysbiosis, and disruption of the immunological homeostasis has been associated with inflammatory diseases [36,37]. We identified taxa most likely to be associated with DSS/AOM-induced carcinogenesis. Enriched abundances of *Clostridiales* and *Staphylococcus aureus* were observed (Figure 3E). *Clostridiales* have been reported to be less abundant in human patients with IBD but enriched specific *Bacteroides* were also reported [38,39]. Increased abundance of *Staphylococcus aureus* was also observed in DSS/AOM samples (Figure 3E). Toxin produced by *Staphylococcus aureus* influences the microenvironment, possibly lipoteichoic acid and peptidoglycan produced by *Staphylococcus aureus*, as Toll-like receptor 2 ligands, such as lipoarabinomannan (LAM), stimulates mature mast cells to cysteinyl leukotriene synthesis and causes inflammation and immune response [40]. *Staphylococcus aureus* might influence the sphingolipid pathway; Alpha-toxin produced by *Staphylococcus aureus* induces inflammatory cytokines via lysosomal acid sphingomyelinase and ceramides and disrupts endothelial-cell tight junctions [40,41].

This notion is supported by our finding, whereby enrichment of microbial metabolic pathways with tumor promoting roles, such as sphingolipid-signaling and lipoarabinomannan biosynthesis pathways, were enriched in fecal samples from tumor bearing DSS/AOM-treated mice. Concomitantly, GSEA of the whole transcriptome of colonic normal appearing mucosal adjacent to tumors revealed enrichments of pathways involved in fatty acid metabolism, oxidative phosphorylation, and the PI3K-Akt-mTOR, TGF-β, and hypoxia pathways. These diverse changes in host-signaling pathways can be attributed to changes in the host microenvironment. Hypoxia due to rapid tumor growth with impaired neovascularization and inflammation resulting from immune cell activation are hallmarks of cancer. Hypoxia-inducible factors control transcriptional adaptation in response to low oxygen conditions, both in tumor and immune cells [42]. 

Considering the interactions between the gut microbes and host-signaling pathways, sphingosine 1-phosphate, a sphingolipid mediator, regulates various cellular functions via high-affinity G protein-coupled receptors, sphingosine 1-phosphate 1-5. The sphingosine 1-phosphate-sphingosine 1-phosphate receptor-signaling system plays important roles in the regulation of complex inflammatory processes [43]. In addition, sphingolipids have become increasingly recognized as important cell mediators in tumor and inflammatory hypoxia. Recent studies have identified acid sphingomyelinase, a central enzyme in the sphingolipid metabolism, as a regulator of several types of stress stimuli pathways and an important player in the tumor microenvironment. Lipoarabinomannan, on the other hand, is a glycolipid that is a toxic factor associated with *Mycobacterium tuberculosis* [44]. Arabinan polymers are major components of the cell wall in *Mycobacterium tuberculosis* and are involved in maintaining its structure, as well as playing a role in host-pathogen interactions. In particular, LAM has multiple immunomodulatory effects [45]. LAM has been shown to scavenge oxidative radicals in macrophages, and it is also known to act on the PI3K-Akt-mTOR pathway in various cells [32]. This response is initiated through the interaction between *Mycobacterium tuberculosis* cell wall surface components, mostly glycolipids, with cells of the innate immune system, particularly macrophages and dendritic cells. The way macrophages and dendritic cells alter their cytokine secretome, activate or inhibit different microbicidal mechanisms and present antigens and, consequently, trigger the T cell-mediated immune response impacts the host immune response [46]. Taken together, the enrichment of the sphingolipid-signaling and LAM biosynthetic pathways found in the bacterial flora of DSS/AOM mice can affect the host hypoxia related microenvironment and immune systems through fatty acid metabolism and the PI3K-Akt-mTOR pathways. In addition, interactions between the sphingolipid-signaling pathway of microbiota and the host oxidative phosphorylation pathway have been discussed; sphingosine 1-phosphate is produced by submembrane sphingosine kinase type I and is suggested to be a key component of the sphingolipid-signaling pathway. In the nucleus or mitochondria, sphingosine 1-phosphate is produced by sphingosine kinase type II, but mitochondrial sphingosine kinase type II affects mitochondrial oxidative phosphorylation. Therefore, we speculated that enrichment of the sphingolipid pathway in the bacterial flora could thereby affect host oxidative phosphorylation. 

With regards to changes observed in DSS- and AOM-treated mice, it was interesting to note that similar expression patterns of differentially expressed genes and OTU microbial composition were observed and differed from both untreated control and DSS/AOM-treated mice. Based functional profiles of differentially expressed genes, various pathways associated with GPCRs were enriched in mucosal samples from these mice. Additionally, pathways associated with type II interferons and chemokine signaling, as well as macrophage signaling were enriched, indicating persistent inflammation instigated by the early exposure to DSS and AOM. It is very likely that the treatments altered the microbial composition of the gut and along with the initial erosion or damage led to long lasting alterations of the colon in these mice. Consistent with this notion is the increased number of lymphoid aggregates still present in the colons of DSS- and AOM-treated mice even weeks after the last treatment. Moreover, type II interferons are usually produced by CD8+ T cells, natural killer cells and natural kill T cells and may be related to enriched type II interferon signature. Nevertheless, while both may be capable of inducing inflammatory changes, neither treatment was able to induce tumor formation, which was the primary focus of our study. 16S rDNA sequencing has been a mainstay for bacterial composition analysis; however, there may be some limitations when it comes to functional profiling. Functional profiles derived from 16S rDNA amplicon sequencing are inferred and are not as robust as those from shotgun metagenomics, thus additional studies may be needed to fully evaluate the biological significance of these findings [47].

## 5. Conclusions

In summary, we have shown that the early inflammatory and carcinogenic effects of DSS and AOM drive inflammation-induced carcinogenesis, disrupting the gut microbiome that, in turn, may act to further promote cancer progression through bacteria-host interactions (Figure 7). Namely, we have revealed enriched sphingolipid signaling and LAM biosynthesis as altered metabolic pathways in DSS/AOM-induced colon cancer. Our findings also suggest an interaction between the PI3K-Akt-mTOR pathway and LAM synthesis in colitis-induced carcinogenesis. Through a comprehensive analysis of the gut microbiota and host tissues we have inferred this interaction. Thus, regulation of gut microbiota could be a potential breakthrough for the prevention and treatment of colorectal cancer.

## Figures and Tables

**Figure 1 cancers-13-03683-f001:**
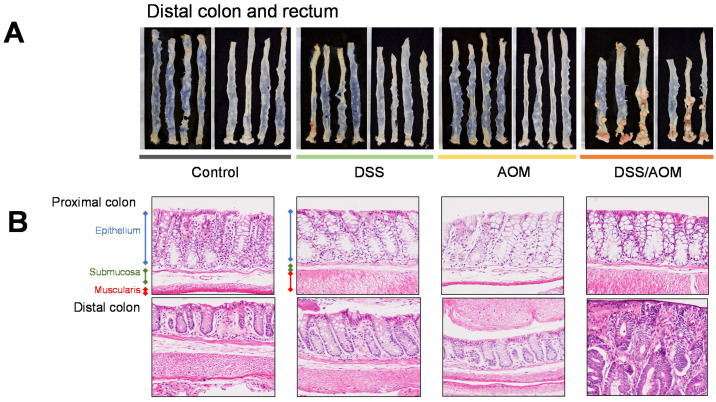
Induction of inflammation induced colon cancer in mice. Female BALB/c were randomized as untreated controls or mice treated with three daily doses of AOM and/or three subsequent cycles of 2% DSS-supplemented for four weeks. (**A**) Gross visualization of colon tumor development six months after the indicated treatment. (**B**) Representative hematoxylin and eosin -stained sections showing histopathological changes in colonic mucosa after the indicated treatments. DSS, dextran sodium sulfate; AOM, azoxymethane; DSS/AOM; dextran sodium sulfate + azoxymethane. High power magnification = 200×, low power magnification = 100×.

**Figure 2 cancers-13-03683-f002:**
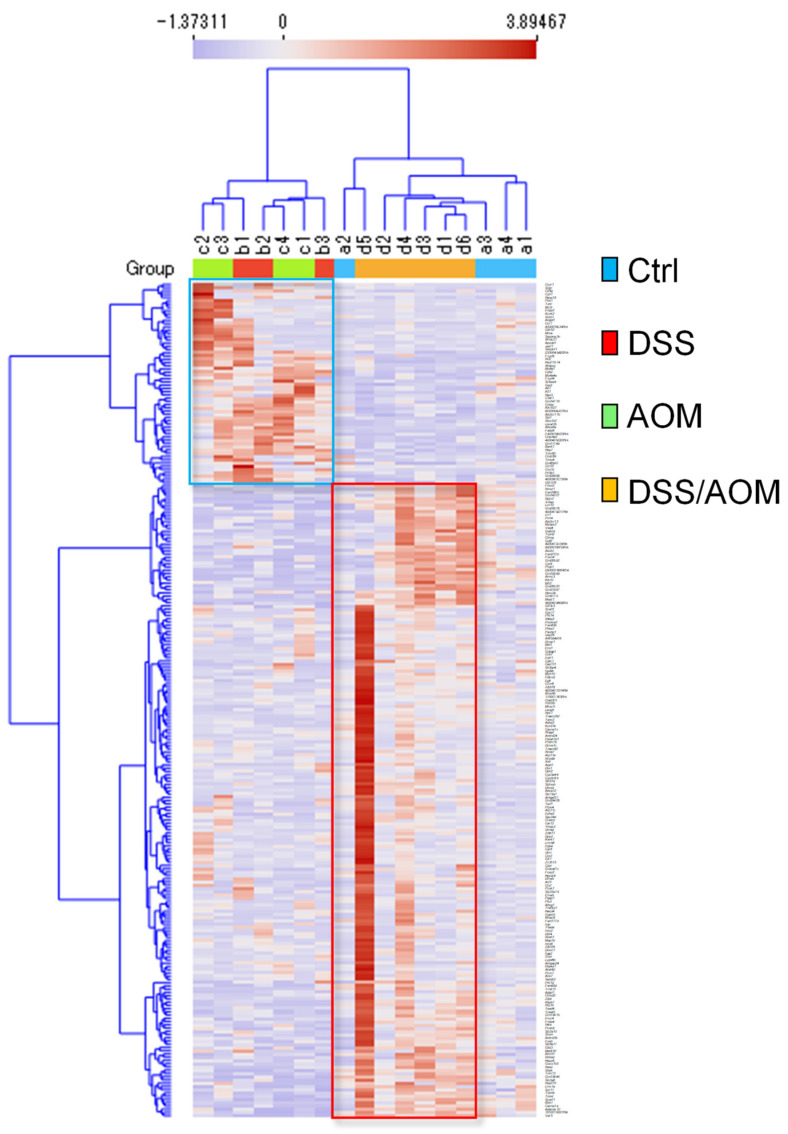
Whole transcriptome analysis of mouse colonic mucosa. Heatmap showing unsupervised hierarchical clustering of genes differentially expressed in DSS/AOM-treated mice. Clustering was based on Euclidean distance and Ward’s linkage. Blue square represents genes upregulated in DSS and AOM mice and red square represents genes upregulated in DSS/AOM mice. Ctrl; control, DSS, dextran sodium sulfate; AOM, azoxymethane; DSS/AOM; dextran sodium sulfate + azoxymethane.

**Figure 3 cancers-13-03683-f003:**
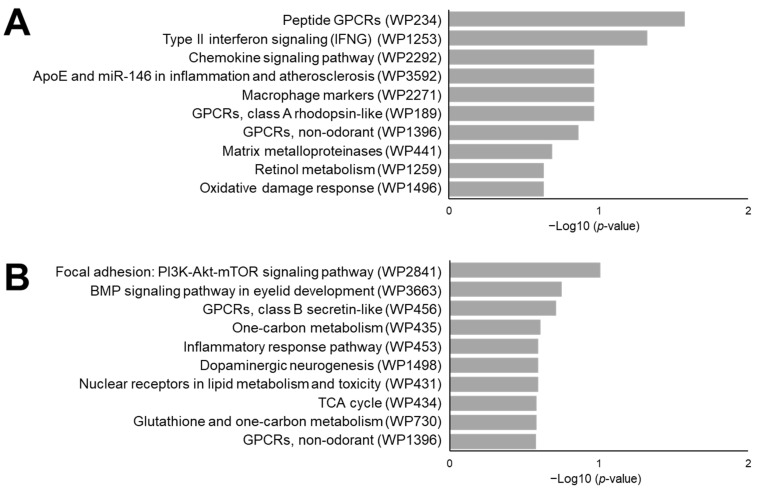
Functional gene expression profiling of colonic mucosa of DSS/AOM mice and control mice. The top ten biological pathways from the mouse 2019 WikiPathways database enriched from genes upregulated in DSS- and AOM-treated mice (blue square in Figure 2) (**A**) and genes upregulated in DSS/AOM-treated mice (red square in Figure 2) (**B**). DSS, dextran sodium sulfate; AOM, azoxymethane, DSS/AOM; dextran sodium sulfate + azoxymethane, GPCRs; G protein-coupled receptors, BMP; bone morphogenetic protein, WP; WikiPathways.

**Figure 4 cancers-13-03683-f004:**
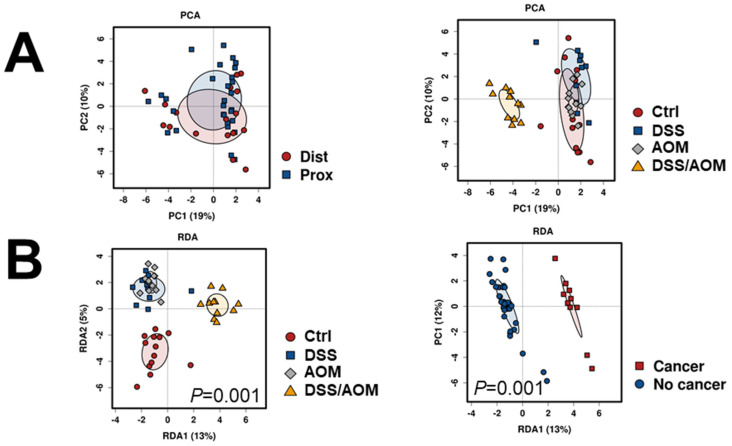
Metagenomic profiling of fecal microbes in mouse DSS/AOM-induced colon cancer. 16S rDNA metagenomic sequencing was performed to generate OTU abundances in proximal (Prox) and distal (Dist) colon fecal samples from control, DSS-, AOM- and DSS/AOM-treated mice. (**A**) Principal component analysis showing dissimilarity between site and treatment. (**B**) Redundancy analysis (RDA) showing significance of treatment cancer as a factor contributing to variation in microbial composition of fecal microbes. (**C**) Hierarchical clustering and heatmap of core taxa (top 300 OTUs). (**D**) Genus-level LEfSe (Linear discriminant analysis Effect Size) analysis showing taxa associated with cancer (DSS/AOM) and no cancer (Control, DSS and AOM). (**E**) Pearson correlation analysis showing the top 25 taxa correlated to treatment. Ctrl; control, DSS, dextran sodium sulfate; AOM, azoxymethane, DSS/AOM; dextran sodium sulfate + azoxymethane, LDA; linear discriminant analysis.

**Figure 5 cancers-13-03683-f005:**
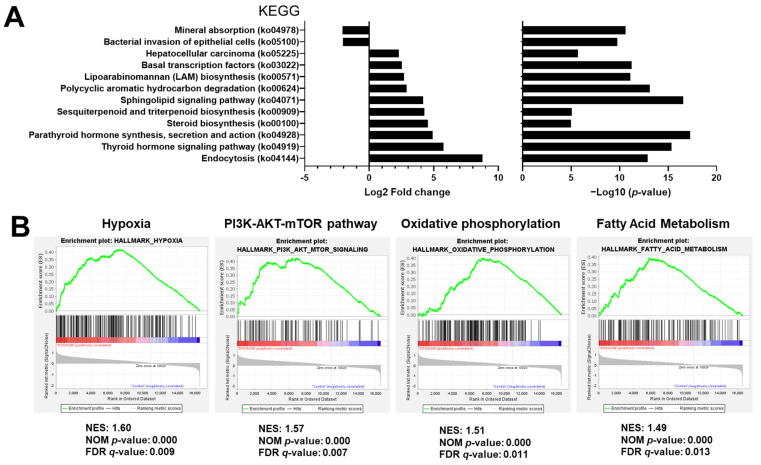
Integrated functional prediction profiling. Piphillin was used to infer and compare functional metabolic pathways from 16S rDNA sequencing abundance data altered in fecal samples from DSS/AOM-treated mice. (**A**) Bar plots showing the fold change (left panel) and the −log10 *p* value (right panel) of the top 12 KEGG pathways enriched in DSS/AOM-treated mice versus controls (Kyoto Encyclopedia of Genes and Genomes, https://www.kegg.jp, access date: 11 May 2021). (**B**) Gene set enrichment analysis showing key related-signaling pathways altered in the normal mucosal of DSS/AOM-treated and control mice. NES, normalized enrichment score; NOM *p* value, nominal *p* value (<0.05 was considered significant); FDR *q* value: false discovery rate (<0.25 was considered significant), DSS/AOM; dextran sodium sulfate + azoxymethane.

**Figure 6 cancers-13-03683-f006:**
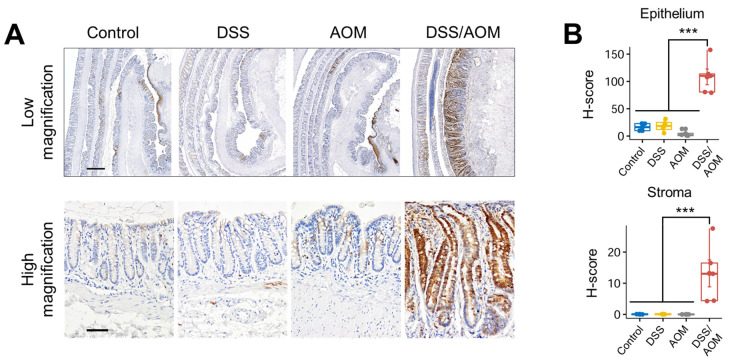
Immunohistochemical analysis of phosphorylated S6 ribosomal protein. (**A**) Representative photomicrographs of phosphorylated S6 immunostaining in the distal colons of control, DSS-, AOM-, and DSS/AOM-treated mice showing low magnification and high magnification fields. Low magnification scale bar = 500 µm and high magnification scale bar = 100 µm. (**B**) H-scores of p-S6 in the epithelium (upper) and stroma (lower) of the distal colonic mucosa. ANOVA *p* value < 0.001 and post hoc statistical analysis was performed using Student-Newman-Keuls Method; *** *p* < 0.001, DSS, dextran sodium sulfate; AOM, azoxymethane, DSS/AOM; dextran sodium sulfate + azoxymethane, ANOVA; analysis of variance.

**Figure 7 cancers-13-03683-f007:**
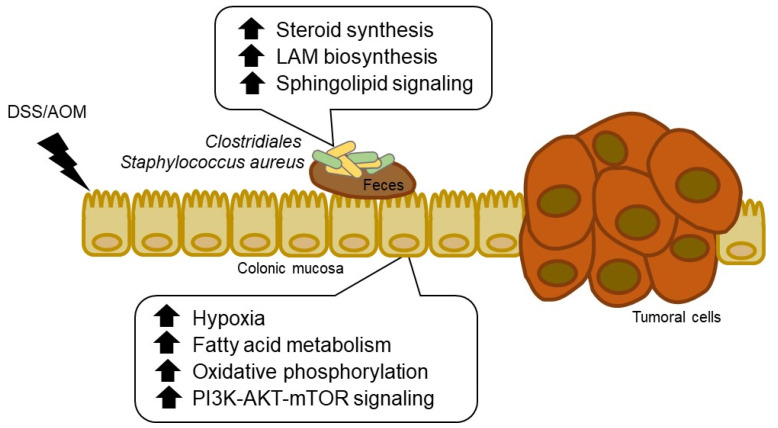
Schematic summary of host transcriptome and microbiome signaling-pathway on mouse DSS/AOM-induced colon cancer. DSS/AOM; dextran sodium sulfate + azoxymethane, LAM; lipoarabinomannan.

**Table 1 cancers-13-03683-t001:** Gene Set Enrichment Analysis (GSEA) results according to the MSigDB Hallmark gene sets.

MSigDB Hallmark Pathway	NES	NOM *p*-Value	FDR *q*-Value
Protein secretion	1.75	0	0.005
TGF-β signaling	1.73	0.001	0.003
mTORC1 signaling	1.7	0	0.005
Cholesterol homeostasis	1.68	0.001	0.006
G2M checkpoint	1.67	0	0.005
UV response down-regulated	1.6	0	0.01
Hypoxia	1.6	0	0.009
E2F targets	1.58	0	0.009
Glycolysis	1.57	0	0.009
Hedgehog signaling	1.57	0.008	0.008
PI3K-Akt-mTOR signaling	1.57	0	0.007
Mitotic spindle	1.55	0	0.009
Interferon-α response	1.54	0.004	0.009
Oxidative phosphorylation	1.51	0	0.011
Reactive oxygen species pathway	1.51	0.008	0.011
Epithelial mesenchymal transition	1.5	0	0.012
Fatty acid metabolism	1.49	0	0.013
UV response up-regulated	1.43	0.002	0.025
Interferon-γ response	1.43	0.002	0.024
Androgen response	1.42	0.013	0.026

NES; normalized enrichment score, NOM; nominal, FDR; false discovery rate.

## Data Availability

The data presented in this study are available on request from the corresponding author.

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
