# Peer review of "Transcriptome Profiling and Metagenomic Analysis Help to Elucidate Interactions in an Inflammation-Associated Cancer Mouse Model"

_cancers, 2021, doi:10.3390/cancers13153683_

Round 1

Reviewer 1 Report

I consider the text presented for review to be very interesting and substantive. 

I have a few minor comments: 

  1. The title of the work should contain a note that the research concerns an animal model - made on mice.
  2. It would be very interesting if the authors would present the weight of the mice at the beginning of the experiment and at the end of the experiment with a description of whether there were any differences in the weight of animals in the study groups(DSS, AOM, DSS/AOM) and in relation to the control group. 
  3. Figure no. 2 is a bit difficult to read. I propose a division into two separate figures; first with heatmap and second containing plots.
  4. Figure no. 3 is also a bit difficult to read. I propose to divide this figure into 3 parts. The first contains the plots with the PCA analysis results, the second contains the heatmap and the third contains the remaining plots. The caption for Figure 3 contains an unnecessary period in the penultimate line. 
  5. Is it possible to give the date of issue of consent to conduct an animal experiment?
  6. Would it be possible to add a graphical (in the form of a model) total presentation of the results of the analyzes performed? Such visualization helps to synthesize the results and makes it more readable. 

Author Response

We thank the reviewer for insightful comments, which we feel have helped us to improve our manuscript. Our specific responses to the points raised are as follows:

  1. The title of the work should contain a note that the research concerns an animal model - made on mice.

RE: We have revised the title to specify a "mouse model".

  1. It would be very interesting if the authors would present the weight of the mice at the beginning of the experiment and at the end of the experiment with a description of whether there were any differences in the weight of animals in the study groups (DSS, AOM, DSS/AOM) and in relation to the control group. 

RE: We have provided the body weight of the mice in supplementary Fig. S1. No changes in mouse bodyweights were noted in the early phase of the experiment, however, mice in the DSS/AOM group experienced gradual bodyweight declines from day 78-112 which remained stable for the duration of the experiment. The data is shown in Supplementary Fig.S1 and have included these comments in the Results section (L164-167).

  1. Figure no. 2 is a bit difficult to read. I propose a division into two separate figures; first with heatmap and second containing plots.

RE: We divided figure 2 into Figure 2 and Figure 3 as suggested.

4-1. Figure no. 3 is also a bit difficult to read. I propose to divide this figure into 3 parts. The first contains the plots with the PCA analysis results, the second contains the heatmap and the third contains the remaining plots. The caption for Figure 3 contains an unnecessary period in the penultimate line. 

RE: We have added higher resolution plots and have reformatted the original Figure 3, now Figure 4 to improve the readability.

4-2. Is it possible to give the date of issue of consent to conduct an animal experiment?

RE: We added the date of approval of animal experiment in the Method section (L88-89)

4-3. Would it be possible to add a graphical (in the form of a model) total presentation of the results of the analyzes performed? Such visualization helps to synthesize the results and makes it more readable. 

RE: We have added the schematic summary of the results as Figure 7 and these explanation is described in the Discussion section (L443).

Reviewer 2 Report

Study by Kazuko Sakai  “Transcriptome profiling and metagenomic analysis help to elucidate interactions in inflammatory-associated cancer” analysis the association of biopsy transcriptome and fecal microbiome in inflammation associated colorectal cancer mouse model. The major points of criticism and correction:

  1. Lack of clear bioinformatic description of the transcriptome data. No clear quality control parameters are listed (neither in the methods part, nor in the results).
  2. Low number of microbiome reads per sample. It would be useful to see rarefication curve in order to see whether the selected threshold was appropriate or more deep sequencing is required.
  3. The in silico analysis of the microbiome functional pathways from 16S rRNA data is under major criticism (PMID: 31521200). Therefore, I would recommend to perform metagenome analysis in order to confirm the results.
  4. The main point of criticism would be that the current results indicate only the associations with cancer process, but the inflammation related processes are set a side. I would suggest to look at the gradual changes on the transcriptome and microbiome level from normal-inflammation-cancer (gradual increase/decrease) and thereby indicate processes and/or bacteria which would be involved in the transition from inflammation to cancer.
  5. Further, I lack the confirmation of the association on the functional level, e.g. western blot, immunohistochemistry etc. indicating the changes of the pathways (e.g. PI3K-AKT-mTOR pathways) detected by this study.
  6. And finally, a lot of misspelling, grammatical mistakes which made reading the manuscript complicated:

Row 149 evaluation confimed the develoment of -> mistakes in words: confirmed, development;

Row 150 adenocarnoma in all mice. histological evaluation or normal mucosa -> mistakes in word: adenocarcinoma, grammatical mistake: evaluation of normal mucosa

Row 175 showed revealed clusters of genes upregualted or downregulaed, -> mistakes in words: upregulated, downregulated; one word showed or revealed should be chosen;

Row 177 mice clustered togethere, -> mistake in word together

Row 207 whether differences gut microbial composition -> mistake: whether differences of gut microbial;

Row 212 between the the different treatmetn -> two time written the; mistake in word treatment

Row 215 to evluate the effect; Row 216 that treatment of DSS/AOM and the the -> two time written the, mistake in word evaluate;

Row 221 treatemnets. Overall, fecal samples -> mistake in word treatments

Row 224 Next, we performed aimed to identify -> grammatical mistake

Row 225 LEfSe analysis revelaed genus-level; -> mistake in word revealed

Row 227 performed correation analysis using; -> mistake in word correlation

Row 231 analysis comparing inflamation induced coloncancer; -> mistake in word inflammation

Row 237 Functiona predictions were inferred; -> mistake in word functional

Row 242 host regulation of hypxia; -> mistake in word hypoxia

Row 247 strucutre which in turn; -> mistake in word structure;

Row 254 Redundancy analysis (RDA) showing RDA significance; -> two time RDA

Row 257 and no cancer (Control, DSS and AOM). level Redundancy analysis (RDA) -> the sentence is not clear.

Author Response

We thank the reviewer for insightful comments, which we feel have helped us to improve our manuscript. Our specific responses to the points raised are as follows:

  1. Lack of clear bioinformatic description of the transcriptome data. No clear quality control parameters are listed (neither in the methods part, nor in the results).

RE: We have added sequencing quality of the whole transcription data including total reads and on target rate in the Method section (L101-105).

  1. Low number of microbiome reads per sample. It would be useful to see rarefication curve in order to see whether the selected threshold was appropriate or more deep sequencing is required.

RE: We added rarefaction curve below. Rarefaction analysis indicated that our sequencing depth was adequate. These results are described in result section (L239-240).

  1. The in silico analysis of the microbiome functional pathways from 16S rRNA data is under major criticism (PMID: 31521200). Therefore, I would recommend to perform metagenome analysis in order to confirm the results.

RE: Thank you for an important recommendation. We understand the reviewer's comments regarding some of the limitations of functional pathway profiling from 16S rRNA amplicon sequencing. Unfortunately, we are unable to perform shotgun metagenomic analysis in the present study. Nevertheless, 16S rRNA sequencing is still a useful tool and is widely used we believe that our current study provides relevant data that could be followed up with additional functional and mechanistic studies and hope to include metagenomic analysis future studies. We have addressed this issue in the discussion section (L433-437).

  1. The main point of criticism would be that the current results indicate only the associations with cancer process, but the inflammation related processes are set a side. I would suggest to look at the gradual changes on the transcriptome and microbiome level from normal-inflammation-cancer (gradual increase/decrease) and thereby indicate processes and/or bacteria which would be involved in the transition from inflammation to cancer.

RE: Thank you for your valuable comments. In this report we focused on the changes in normal appearing mucosa of DSS/AOM-induced tumor bearing mice. Based on the assumption that this mucosa is predisposed to develop tumors, one could infer that these may be some factors associated with colon carcinogenesis. However, we must also recognize that these changes that are not only influenced by local factors but also by regional and to some degree systemic additional factors. As the reviewer points out it would be worthwhile to investigate temporal changes at the transcriptome and microbiome level and is a project that we are working on. Unfortunately, these models take some time to develop and selecting key time points is essential. In addition, we plan to also characterize immune cell component in addition to identifying potential biomarkers and this study will serve as the foundation for how conduct additional studies investigating the temporal changes in the early stages of DSS/AOM-induced colon carcinogenesis.

  1. Further, I lack the confirmation of the association on the functional level, e.g. western blot, immunohistochemistry etc. indicating the changes of the pathways (e.g. PI3K-AKT-mTOR pathways) detected by this study.

RE: To confirm of the association on the functional level, we have included immunohistochemistry staining and quantification of phosphorylated S6 ribosomal protein (p-S6) in the distal colons of mice. p-S6, which is a downstream signal molecule the of PI3K-AKT-mTOR pathway was strongly expressed in the epithelium and stroma of DSS/AOM group compared with other groups and was statistically significant when quantified using the H-score as a metric. Thus, we could successfully confirm the changes of PI3K-AKT-mTOR pathways at the functional levels. These results are included the results section (L308-319) and Figure. 6). Methods for IHC staining and quantification are included in the Methods section 2.5 section (L142-154)

And finally, a lot of misspelling, grammatical mistakes which made reading the manuscript complicated:

RE: We are very sorry for such a troublesome task. We have corrected spelling and grammatical errors throughout the manuscript.

Round 2

Reviewer 2 Report

Thank you for the thorough replies. Addition of immunohistochemistry significantly improved the manuscript. I would still suggest to go through the manuscript thoroughly, as I still noticed multiple spelling, typing errors. 

And I would suggest to change one word in the title - inflammation instead of the inflammatory.

Author Response

Response to Reviewer #2

We would like to thank you for your suggestions to improve our manuscript.

RE: Thank you for the thorough replies. Addition of immunohistochemistry significantly improved the manuscript. I would still suggest to go through the manuscript thoroughly, as I still noticed multiple spelling, typing errors.

And I would suggest to change one word in the title - inflammation instead of the inflammatory.

We have changed the word ‘inflammatory’ to ‘inflammation’ in the title accordingly. We apologize for the remaining multiple spelling mistakes. We have corrected spelling and grammatical errors throughout the manuscript. In addition, we have added abbreviations to the text.